# Effect of Carbohydrate-Electrolyte Solution Including Bicarbonate Ion Ad Libitum Ingestion on Urine Bicarbonate Retention during Mountain Trekking: A Randomized, Controlled Pilot Study

**DOI:** 10.3390/ijerph18041441

**Published:** 2021-02-04

**Authors:** Masahiro Horiuchi, Tatsuya Hasegawa, Hiroshi Nose

**Affiliations:** 1Division of Human Environmental Science, Mount Fuji Research Institute, Yamanashi 4030005, Japan; haset@mfri.pref.yamanashi.jp; 2Department of Sports Medical Sciences, Shinshu University Graduate School of Medicine, Matsumoto 3908621, Japan; nosehir@shinshu-u.ac.jp

**Keywords:** acute mountain sickness, arterial hypoxemia, bicarbonate buffering system, dehydration, heart rate, O_2_ pulse

## Abstract

We investigated whether bicarbonate ion (HCO_3_^−^) in a carbohydrate-electrolyte solution (CE+HCO_3_) ingested during climbing to 3000 m on Mount Fuji could increase urine HCO_3_^−^ retention. This study was a randomized, controlled pilot study. Sixteen healthy lowlander adults were divided into two groups (six males and two females for each): a tap water (TW) group (0 kcal with no energy) and a CE+HCO3 group. The allocation to TW or CE+HCO3 was double blind. The CE solution contains 10 kcal energy, including Na^+^ (115 mg), K^+^ (78 mg), HCO_3_^−^ (51 mg) per 100 mL. After collecting baseline urine and measuring body weight, participants started climbing while energy expenditure (EE) and heart rate (HR) were recorded every min with a portable calorimeter. After reaching a hut at approximately 3000 m, we collected urine and measured body weight again. The HCO_3_^−^ balance during climbing, measured by subtracting the amount of urine excreted from the amount of fluid ingested, was −0.37 ± 0.77 mmol in the CE+HCO3, which was significantly higher than in the TW (−2.23 ± 0.96 mmol, *p* < 0.001). These results indicate that CE containing HCO_3_^−^ supplementation may increase the bicarbonate buffering system during mountain trekking up to ~3000 m, suggesting a useful solution, at least, in the population of the present study on Mount Fuji.

## 1. Introduction

Mountain trekking consists of prolonged exercise with several periods of rest at high-altitude. For practical implication to mountain trekking, effective nutritional strategies in response to both prolonged and hypobaric hypoxic exercise may be required.

Prolonged exercise requires more energy and affects the dehydration status. Dehydration status may cause early occurrence of lactate threshold, indicating early onset of fatigue during exercise [1,2]. Acute hypoxic exercise further increases blood lactate concentrations compared to normoxic exercise [3,4]. It is well known that accumulating lactate in active tissues can be buffered by bicarbonate buffering system [5].

In this regard, at high altitudes (hypobaric hypoxia), several studies have investigated the effects of carbohydrate (CE) or bicarbonate ion (HCO_3_^−^) supplementation on exercise performance and/or physiological responses, and resulted in controversial findings. For example, time trial performance improved [6,7], whereas the same groups also found that time trial performance did not improve time trial performance [8]. Some laboratory studies also found positive effects of bicarbonate ion (HCO_3_^−^) ingestion on anaerobic exercise performance in hypoxia [9,10,11,12,13], but a few studies reported no effect of HCO_3_^−^ ingestion on exercise performance in hypoxia [14,15]. While these previous studies examined effects of CE or HCO_3_^−^ [11,12,13,14,15,16,17], separately, one study investigated the combined effects of CE and HCO_3_^−^ supplementation on sprint performance in normoxia, and found no improvement in sprint performance [16]. However, mountain trekking may not require maximal anaerobic capacity (e.g., time trial performance). More importantly, in these previous studies [11,12,13,14,15,16,17,18], a large amount of HCO_3_^−^ was supplied pre-exercise. Under these conditions, it has been reported that acute gastrointestinal distress is a known side-effect of ingesting large amounts of HCO_3_^−^ [17]. Furthermore, it has been suggested that gastrointestinal distress may restrain individuals from using HCO_3_^−^, regardless of its potential ergogenic benefits [18]. Therefore, symptoms such as diarrhea and/or vomiting may indicate a major practical limitation for trekkers, suggesting ad libitum supplementation based on individual feeling might be better tactics during prolonged mountain trekking.

As mentioned previously, prolonged mountain trekking at high altitude may cause dehydration and accumulation of lactate, and hence, renal regulation of the acid–base balance and body fluid distribution is particularly important at high altitudes and seems to play a significant role in the development of acute mountain sickness (AMS) [19,20]. Indeed, at high altitudes, elevation in the renal excretion of HCO_3_^−^ was observed compared to that at low and moderate altitudes [21]. Urine sampling has several advantages over blood sampling. As the hygiene study condition is important at actual fields, urine sampling may be useful compared with the invasive blood sampling technique. In clinical settings, standard tests exist to assess renal function such as a calculation of renal clearance [22]; however, these tests require anaerobic arterial blood and measurements of urine flow, and are thus are difficult to employ in the context of high altitude.

Accordingly, the aim of this study was to examine the effects of CE+HCO3 solution supplementation on urine HCO_3_^−^ retention during mountain trekking at high altitude, and secondarily, on AMS. It was hypothesized that CE+HCO3 solution supplementation would increase urine HCO_3_^−^ retention and provide potential benefits to prevent AMS.

## 2. Materials and Methods

All the procedures in this study were approved by the institutional ethical committee at Mount Fuji Research Institute according to the guidelines of the Declaration of Helsinki (ECMFRI-02-2014).

### 2.1. Study Location

As a field for this study, Mount Fuji in Japan was chosen, because more than 250,000 people have visited the mountain annually since 2013, when it was appointed as a World-Heritage Site. In this study, two huts were provided for pre- (2305 m) and post- (3000 m) measurements of mountain trekking. The measurements were performed in a sufficient space in the living room, and the room temperature was controlled at approximately ~22 °C. The climbing road starts at the 5th station (2305 m) and is well maintained. The road surface is mainly covered by gravel or sand. Climbers are not required to have any special techniques, such as using an ice axe, and thus, beginners can climb.

### 2.2. Participants

After a detailed explanation of all study procedures, including the possible risks and benefits of participation, each participant gave his written consent (Figure 1). Sixteen healthy adult participants were randomly divided into two groups: the tap water (TW) group and the CE solution containing HCO_3_^−^ (CE+HCO_3_) group. The participants were recruited for the present study through an advertisement at the Mount Fuji Research Institute and local area of Fuji-Yoshida city and Kawaguchiko town. The recruitments were performed from 8 April to 26 June, 2016 (n = 29). As shown in Figure 1, applicants who were free from any known known cardiovascular or cerebrovascular diseases and had not taken any medications were enrolled (n = 16). Additionally, none of the participants was exposed to an altitude higher than 1500 m within 6 months before the study, and they were confirmed to be without a prior history of symptoms of AMS [23]. The allocation to TW or CE+HCO_3_ was double-blind; experimenters learned of participant’s allocation only after the statistical analysis had been performed on all outcome measures. This randomized allocation was conducted by a staff-member not directly involved with the mountain trekking study, and was stratified by sex and age. The allocation ratio to each group was 1:1 (n = 8 for each group). Participants were requested to abstain from caffeinated beverages for 12 h and from strenuous physical activity and alcohol for at least 24 h before the study [23]. Their physical characteristics and baseline values are given in Table 1.

### 2.3. Beverage and Foods

Before climbing, the participants were provided with three 500 mL bottles. For the TW group, the bottle contained 0 kcal energy and 0 g nutrients. For the CE+HCO_3_ group, the bottles contained the following per 100 mL, totaling 10 kcal of energy: carbohydrate, 2.5 g (glucose, 1.8 g); protein, 0 g; fat, 0 g; Na^+^, 115 mg; K^+^, 78 mg; H_2_PO_4_^2−^, 6.2 mg; Mg_2_^+^, 2.4 mg; Cl^−^, 177 mg; and HCO_3_^−^, 51 mg. Similarly, both groups were provided with the same food menu (cereal and energy bar, and chocolate) before climbing, with a total energy of 961 kcal, and containing carbohydrate 131 g, protein 14.7 g, fat 41.8 g, and Na 417 mg.

Notably, the participants were only provided either TW or CE+HCO_3_ beverage, and nutrient composition and the main aim (i.e., effects of different drinks on physiological responses and symptoms of AMS) of this study were not revealed to the participants. Thus, the participants could know only their own taste [24].

### 2.4. Procedure

The participants came to the parking area at 2305 m by car at approximately 13:30 after a light lunch (provided same sandwich) about 1 h prior. After emptying their bladders, participants were measured for body weight including their clothes and boots within a precision of 50 g (UC-321, A&D Instruments, Tokyo, Japan). Their body weight without clothing to determine the total weight of their clothing, boots, and backpack (weighing ~7 kg and containing a jacket, a sweater, food, and the bottles) (the difference in the two measured body weights) was also measured. To estimate their VO_2peak_, the participants underwent a graded walking test without the backpack in a flat space of the parking area as detailed below. The participants started climbing at approximately 15:00 by their own pace, during which time they were allowed to take the food and beverages ad libitum, and energy expenditure (EE) and heart rate (HR) were continuously measured. After reaching a hut at approximately 3000 m, we measured their body weight without clothing again before dinner after collecting their urine samples. The samples were used to measure their volumes at the hut and approximately 50 mL of each sample was stored in a sample tube to measure the composition in a laboratory thereafter (Figure 2).

### 2.5. Estimation of Peak Oxygen Uptake

The graded walking test consisted of subjectively slow, moderate, and fast speeds of walking for 3 min each, during which EE was measured in each 3 min period with a portable calorimeter (JD Mate; Kisseicomtec, Matsumoto, Japan) and HR was measured with a portable HR monitor (RS 800CX; Polar Electro Japan, Tokyo, Japan). The EE was obtained after converting the signals from a tri-axial accelerometer and a barometer in the device to the oxygen consumption rate (VO_2_) according to the previously reported equation [25]. VO_2peak_ and peak HR (HR_peak_) at the VO_2peak_ were determined from the averaged values of the last three consecutive values from the fast walking period. When HR_peak_ was lower than the age-predicted maximal HR in participants, we estimated VO_2peak_ by extrapolating the VO_2_ value at the age-predicted maximal HR [26] according to a regression equation between HR and VO_2_ during the graded walking test.

### 2.6. Measurements during Trekking

Exercise intensity with the portable calorimeter (JD Mate) and HR with the HR monitor (RS 800CX) were measured every min during climbing. The energy and nutrition obtained from food ingested during climbing were recorded and analyzed thereafter using a software package (Excel Eiyo-kun; Kenpakusha Co., Ltd., Tokyo, Japan). The total volume of the beverages consumed during climbing was recorded at the hut based on the volume that remained in each bottle. The periods when VO_2_ was less than 15% of VO_2peak_ were regarded as resting periods, as previously described [24,27]. Peripheral arterial oxygen saturation (SpO_2_) was measured by a finger pulse oximeter, and nadir values were used.

The urine samples brought back to the laboratory were stored in a refrigerator at −80 °C for further measurements of HCO_3_^−^ concentration using the titration method and other electrolyte concentrations using standard ion sensitive electrode methods. Regarding the measurement of HCO_3_^−^ concentration in urine using the titration method, the urine sample of individual participants was diluted with three times as much distilled water, and 5 mL of the diluted sample was moved to another vehicle. The vehicle was poured with diluted HNO_3_ (0.01 *N*) or NaOH (0.01 *N*) solution at a constant rate of 1 mL min^−1^ using a non-pulsatile pump for the liquid chromatography, while pH was continuously measured with a pH electrode. From the titration curve showing the relationship between pH and the total poured volumes of HCO_3_ or NaOH solutions (Figure 3), HCO_3_^−^ concentration in the sample was determined from the poured volumes of HNO_3_ or NaOH solution from the pH value (6.1) of H_2_CO_3_ at 37 °C. Then, the balance of HCO_3_^−^ in the body during climbing was calculated by subtracting the total loss into urine from the total gain with the CE+HCO_3_ solution. Similarly, the balance of Na^+^ in the body during climbing was also determined. Other urine variables (urine pH, specific gravity and Na^+^) were evaluated by a commercial laboratory service (SRL Co., Ltd., Tokyo, Japan).

### 2.7. Acute Mountain Sickness Assessment

Symptoms of AMS were evaluated using “The 2018 Lake Louise Acute Mountain Sickness Score”. AMS was defined as the presence of a headache and at least one of the following symptoms: gastrointestinal upset (i.e., anorexia, nausea, or vomiting), fatigue or weakness, and dizziness or lightheadedness with total score ≥3 [28]. All participants were asked to respond in terms of the worst symptoms of AMS experienced during climbing.

### 2.8. Sample Size

A sample size estimation for the primary analysis (urine HCO_3_^−^) indicated that approximately 6 participants for each group were needed to produce an 80% chance of obtaining statistical significance at the 0.05 level (G Power 3.1). This required sample size was calculated based on a previous study that a minimum significant change of urine HCO_3_^−^ between sea level and high altitude >3000 m is ~5 mmol L^−1^ [29]. Eight participants for each group completed the experimental protocol to account for potential missing data.

### 2.9. Statistical Analyses

A statistical software package was used for all the analyses (Sigma Stat ver. 3.5, Hulinks, IL, USA). An unpaired t-test was used to examine the significant differences between the TW and CE+HCO_3_ groups for the following variables: physical characteristics, estimated VO_2peak_, SpO_2nadir_, walking and resting time, average EE, HR, and EE/HR, and the HCO_3_^−^ and Na^+^ balance. A two-way repeated measured ANOVA was used to compare the body weight changes and urine pH pre- and post-trekking. A rank scale variable such as symptoms of AMS was compared between two groups by the Mann–Whitney U test. The values are shown as means ± standard deviation for the eight participants in each of the TW and CE+HCO_3_ groups. A *p* value less than 0.05 was considered statistically significant.

## 3. Results

All of the 16 participants (eight for each group) successfully completed mountain trekking, and there were no missing data. Therefore, all data were used for further analysis. Due to very bad weather conditions (i.e., rainy and windy) after reaching the hut, the study was terminated after staying overnight at the hut. There were no significant differences in physical characteristics, HR, baseline SpO_2_, and estimated VO_2peak_ between two groups (all *p* > 0.05), whereas SpO_2nadir_ in the TW was significantly lower than CE+HCO_3_ (*p* = 0.022, Table 1).

Figure 2 shows a typical example of VO_2_, HR, and altitude during mountain trekking in a male participant aged 26 years of age in the TW group. He repeated several bouts of exercise interspersed with several rests for 150 min before reaching the hut. No differences in the volume of beverage consumption between the groups (371 ± 155 mL in the TW vs. 358 ± 177 mL in the CE+HCO_3_, *p* = 0.879). While the TW group did not intake any energy and nutrients from the beverage, the CE group ingested 8 g (6 g) of carbohydrate (glucose), 17.9 meq of Na^+^, and 3.0 meq of HCO_3_^−^ on average.

Table 2 shows a summary of the variables measured during the mountain trekking. There were no statistically significant differences in the following variables, i.e., walking and resting time, averaged EE during walking; however, the average HR and EE/HR during walking in the CE+HCO_3_ group were marginally lower compared with the TW group (*p* < 0.10).

Although no differences in body weight, Na^+^, and HCO_3_^−^ concentration in the TW and CE+HCO_3_ groups (despite the different beverage intakes) were found, there was a significant main effect of time in these variables (Table 3). No effect of drink and time on urine pH was found. The total urine volume during climbing was 330 ± 105 and 343 ± 88 mL in the TW and CE+HCO_3_ groups, respectively, with no significant differences (*p* > 0.05). When the Na^+^ and HCO_3_^−^ ion balances in the body were calculated from the differences in the volumes between the gains from beverage intake and loss as urine, they were 3.61 ± 8.27 (Na^+^) and −0.37 ± 0.77 (HCO_3_^−^) meq, respectively, in the CE+HCO_3_− group, and these values were significantly higher than in the TW group (−16.16 ± 4.74 [Na^+^] and −2.23 ± 0.96 [HCO_3_^−^] meq, respectively; both *p* < 0.001), as shown in Figure 4.

Acute mountain sickness was detected in two of eight participants with TW, defined as “Lake Louise Acute Mountain Sickness Score” of 3 or higher with headache, whereas no AMS participants with CE+HCO_3_. The values of the “Lake Louise Acute Mountain Sickness Score” in the TW were 7, 3, and 2 (n = 1, respectively), 1 (n = 3), and 0 (n = 2). In the CE+HCO_3_, the scores were 1 (n = 2) and 0 (n = 6), respectively. The Mann–Whitney U test found a marginally differences between the two groups (*p* = 0.070).

## 4. Discussion

The major findings in the present study were that the ingestion of CE-containing HCO_3_^−^ solution (1) increased the urine HCO_3_ retention in the body and (2) marginally lowered average HR and AMS scores.

Although the detailed mechanisms of the increased urine HCO_3_ retention in the body with the CE+HCO_3_ ingestion remains unknown, our results can give useful insights for climbers at high altitude. It has been theoretically suggested that bicarbonate ingestion (e.g., NaHCO_3_) may increase the availability of blood HCO_3_^−^ and strengthen the buffering capacity, which acts to dampen the rate of H^+^ accumulation during exercise [30]. Traditionally, large amounts of HCO_3_^−^ in the blood, exceeding 24–28 mM, may be discharged into the urine [31]. Indeed, a previous study showed that urine HCO_3_^−^ markedly increased from the rest to after exercise [32], which is consistent with our results. At high altitude, initial compensatory physiological response is an increase in pulmonary ventilation to deliver sufficient oxygen into peripheral tissues [33]. This increase in ventilation elevates exhaled CO_2_ levels, and thus, a reduction in PCO_2_ at peripheral tissues was observed [33]. Moreover, hyperventilation-induced H^+^ removal was increased, and bicarbonate concentration is also further regulated by renal compensation, the process by which the kidneys regulate the HCO_3_ concentration by secreting H^+^ ions into the urine within a few hours when one reached at high altitude [34]. Importantly, increases in H^+^ decrease myofibrillar Ca_2_^+^ sensitivity, and further accelerates muscle fatigue in the presence of a decreased Ca_2_^+^ transient amplitude [35]. It has also been reported that accumulation of H^+^ in skeletal muscle decreases maximal velocity of shortening, leading to a decrease in maximal power output [36]. Given these results, although detailed mechanisms are uncertain and each individual were not forced to perform mountain trekking at their maximal effort, our results show that greater HCO_3_^−^ retention in the urine with CE+HCO_3_ beverage ingestion could provide useful information for mountain trekkers on Mount Fuji.

It should be noted that a previous study demonstrated that would be greater inter-individual variability in extracellular peak blood alkalosis (i.e., responder vs. non-responder), which ranges from 30 to 180 min [37]. While previous laboratory studies ingested NaHCO_3_^−^ pre-exercise only [9,10,11,12,13,14,15], participants in the present study could take CE with HCO_3_^−^ ad libitum. Although highly speculative, as our study protocol lasted ~150 min, ad libitum ingestion of HCO_3_^−^ throughout climbing at high-altitude might be effective to increase bicarbonate buffering system. An important issue is that ad libitum ingestion may avoid side-effects, i.e., gastrointestinal distress, and hence could be more practical for individuals during mountain trekking. However, it should also be note that it was not recorded when and how each individual consumed energy food and/or beverage during ascending, suggesting that optimal strategy of bicarbonate ion ingestion during mountain trekking at high altitude is still unclear.

In the present study, marginally lower average HR values during mountain trekking with CE+HCO_3_ ingestion were found. It has been speculated that the acceleration of body fluid recovery with carbohydrate electrolyte solution could contribute to attenuating an increase in HR during climbing mountain, perhaps, due to maintain stroke volume [24]. Although no differences in body weight changes between the groups were observed, the EE (VO_2_)/HR as an indirect indicator of stroke volume [38] also demonstrated marginally higher values with CE+HCO_3_. These results may suggest that CE+HCO_3_ ingestion could also be affective to accelerate body fluid recovery, leading to a maintenance of stroke volume, and resulting in an attenuation in the HR increases.

Symptoms with AMS in the TW were found in two participants, whereas no participants with AMS were found in in the CE+HCO_3_. Numerous factors such as age [39,40], sex [39], prior history of AMS [41], rapid ascend [42], arterial hypoxemia [43,44], cardiorespiratory responses to hypoxia [41,45], and hydration status, including drinking habits [45,46], which may all contribute to the development of AMS; therefore, it is very difficult to detect a cause of AMS by a single factor. Nonetheless, there may be several possibilities to account for the present results. As mentioned previously, hyperventilation which was observed initially at a high altitude [33] could potentially increase SpO_2_ during a sojourn at high altitude [33]. Arterial hypoxemia has been suggested to be one of the robust candidates to cause AMS [43,44]. If the hypothesis which advocates hyperventilation-induced increases in SpO_2_ as potentially attenuating the severity of symptoms of AMS is taken into account, our results may be reasonable. Although further studies measuring pulmonary ventilation and end tidal CO_2_ might be needed, these measurements in the actual field (i.e., during mountain trekking at high altitude) are almost impossible. Another explanation may relate to different calorie intake between the two groups, as a previous study reported that reduced energy intake after rapid ascent to high altitude is associated with AMS severity [47]. These diet effects on AMS should also be expanded in future study.

### 4.1. Methodological Considerations

There are several limitations to our results. First, our hypothesis was tested for different participant groups, so the effect of different groups on physiological responses could not be completely ruled out. However, as it was impossible to control environmental conditions on the field study, it was decided to conduct this study for different groups on the same day. Furthermore, since there are no differences in physical characteristics and cardiorespiratory variables, our main conclusions may not be strongly affected. Similarly, individual different adaptions to high-altitude, e.g., prevention of symptoms of acute mountain sickness, should be considered. Second, the relatively small sample size should be considered. Thus, post hoc power analysis for pairwise comparisons that were observed with significant differences as the standard of 80% power with a two-sided significance level of 0.05 (G Power 3.1) was conducted. As a result, the estimated effect size of Cohen’s d was 2.137, with (1-β) power of 0.978 for urine HCO_3_^−^ retention and Cohen’s d was 2.933 with (1-β) power of 0.998 for Na^+^ retention in the body. These numbers may be sufficient higher values to detect significant differences in these variables between the TW and CE+HCO_3_ groups. Similarly, the analysis of results was combined with both men and women. Because of the very small sample size (six men and two women for each group), it was impossible to conduct a separate analysis in women and men. However, some previous studies demonstrated that the cardiorespiratory response at high altitude was different between men and women [48,49]. Third, this study consisted of two experimental conditions, i.e., TW and CE+HCO_3_, and thus, the condition of CE or HCO_3_ in isolation could not be done. These study conditions should also be considered because it is still unclear whether carbohydrate or bicarbonate ions have more of a dominant effect on physiological variables. Future research directions may also be highlighted.

### 4.2. Perspectives

Given that greater dose of HCO_3_^−^ may cause acute gastrointestinal distress [17] and bicarbonate buffering system may be more required under hypoxic exercise rather than normoxic exercise [3,4], our results may provide important practical advice for future field investigations in this research area. The benefits of our results will be applied to not only mountain trekking, but also other sports activities at high altitude (e.g., ball games, running, and cross-country skiing).

## 5. Conclusions

Carbohydrate containing HCO_3_^−^ ad libitum ingestion significantly increased urine HCO_3_^−^ retention in the body. These results suggest that CE containing HCO_3_^−^ supplementation may increase the bicarbonate buffering system during mountain trekking ~3000 m. Thus, the ingestion of CE solution containing HCO_3_^−^ might be a useful solution, at least in the population of the present study on Mount Fuji.

## Figures and Tables

**Figure 1 ijerph-18-01441-f001:**
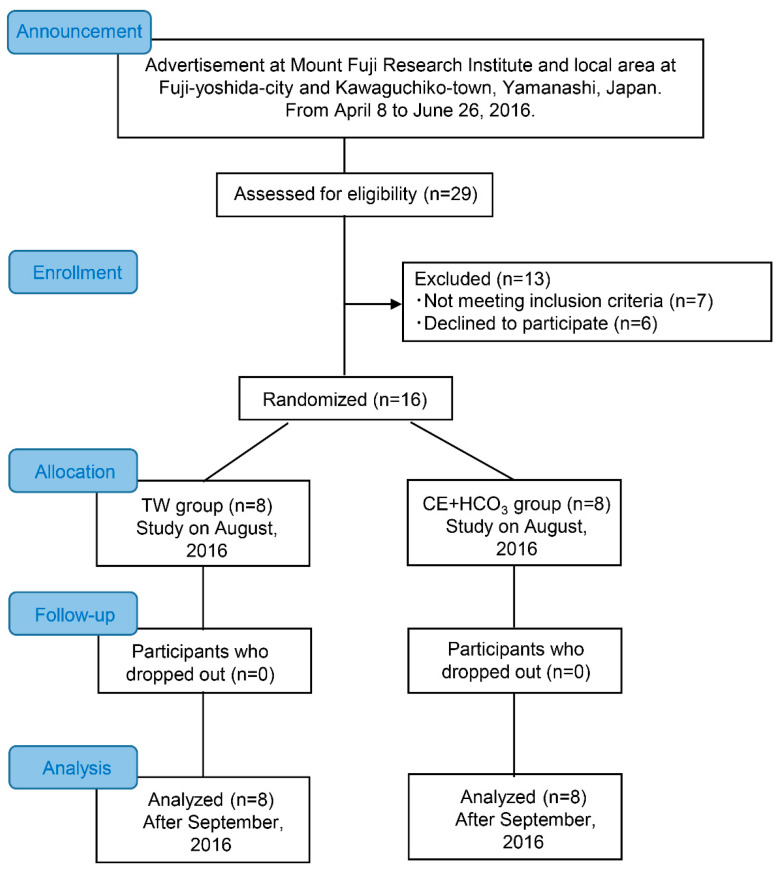
A timeline of the present study. TW, tap water; CE+HCO_3_, carbohydrate solution including bicarbonate ion.

**Figure 2 ijerph-18-01441-f002:**
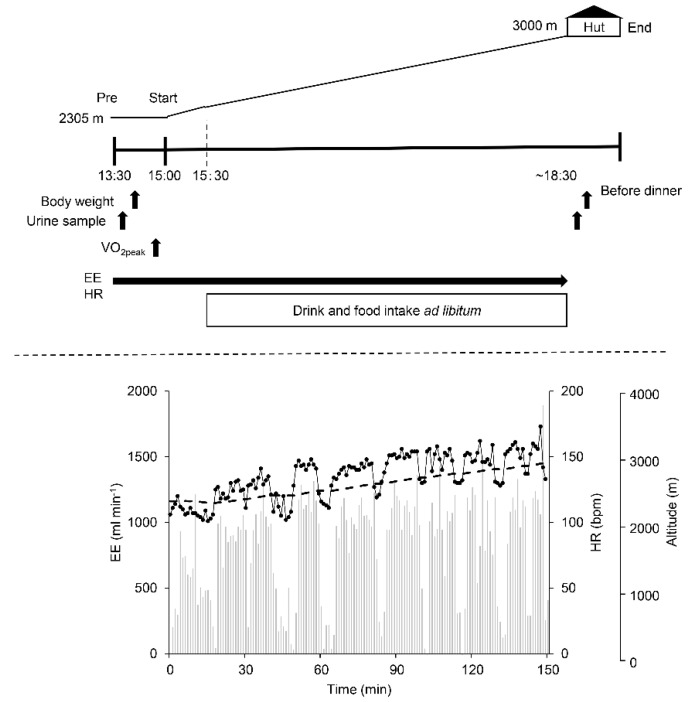
Upper panel: the profile of this study. Participants consumed TW or CE+HCO_3_ beverage ad libitum throughout the mountain trekking except for the first 30 min before beginning the trekking, during which time their body weight and peak aerobic capacity were measured. The measurements are shown in the procedure of the text. Lower panel: typical examples of heart rate (HR), energy expenditure per min (EE) and altitude. Black line graph indicates HR, gray bar graph indicates EE, and gray dotted line indicate altitude.

**Figure 3 ijerph-18-01441-f003:**
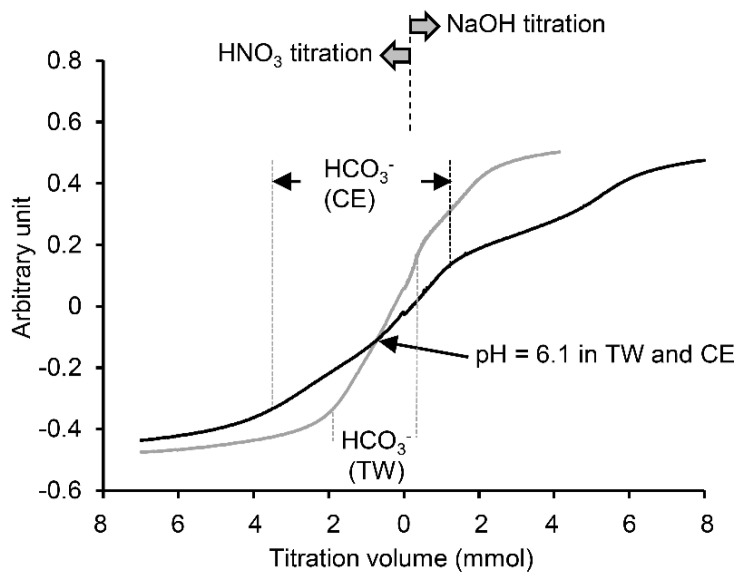
An original recording of urine titration curve for a typical single subject in the tap water (TW; gray line) and carbohydrate solution (CE+HCO_3_; black line) group, respectively. White circles indicate the breaking points for each titration curve and a gray circle indicates pH = 6.1.

**Figure 4 ijerph-18-01441-f004:**
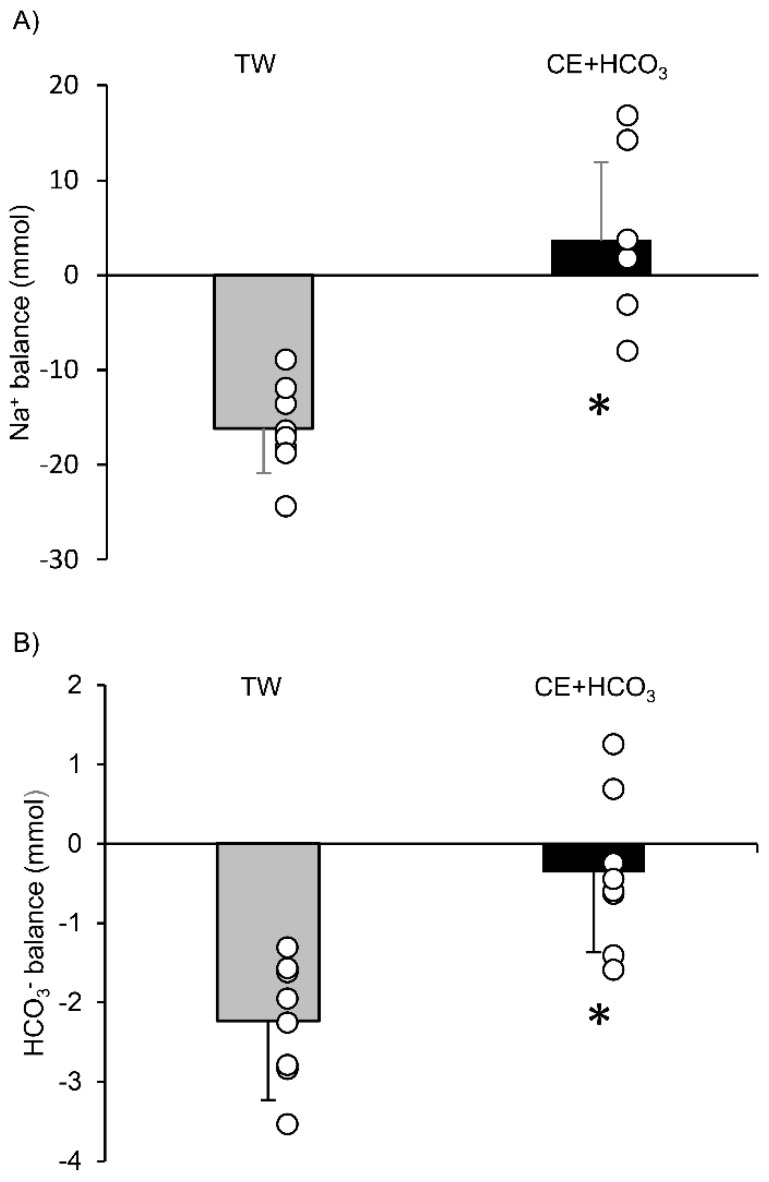
Mean values with SD (bar graph) and an individual data (white circles) of urine Na^+^ (**A**) and HCO_3_^−^ (**B**) balance at the hut between the groups are shown. Gray bars indicate the TW and black bars indicate the CE group. * indicates significant differences between the TW and CE+HCO_3_ groups.

**Table 1 ijerph-18-01441-t001:** Physical characteristics, heart rate, estimated VO_2peak_, and SpO_2nadir_ in the two groups.

	TW	CE+HCO_3_	*p* Values
	(6 M and 2 W)	(6 M and 2 W)	
Age, years	34 ± 9	33 ± 10	0.794
Height, cm	169 ± 7	170 ± 7	0.730
Body weight, kg	61.7 ± 8.1	62.8 ± 9.9	0.799
BMI, kg·(m^2^)^−^^1^	21.6 ± 2.2	21.7 ± 3.0	0.930
HR at standing rest, bpm	80 ± 2	78 ± 4	0.391
HR_peak_ during 9 min walking, bpm	159 ± 15	155 ± 9	0.509
Estimated VO_2peak_, mL·kg^−1^·min^−1^	39.3 ± 9.0	41.7 ± 13.1	0.676
SpO_2_ at rest, %	94 ± 2	95 ± 1	0.862
SpO_2nadir_, %	83 ± 3	87 ± 3	0.022

Values are mean ± standard deviation (SD). TW, tap water; CE+HCO_3_, carbohydrate with bicarbonate ion; M, man; W, woman; BMI, body mass index; HR, heart rate; bpm, beats per minute; VO_2_, pulmonary oxygen uptake; SpO_2_, peripheral arterial oxygen saturation.

**Table 2 ijerph-18-01441-t002:** Summarized results of time and physiological variables during mountain trekking.

	TW	CE+HCO_3_	*p* Values
	(6 M and 2 W)	(6 M and 2 W)	
Walking time, min	121 ± 3	118 ± 3	0.104
Resting time, min	32 ± 5	31 ± 2	0.618
*During walking*			
Average EE, mL·min^−1^	880 ± 130	951 ± 179	0.379
Average HR, mL·min^−1^	138 ± 10	128 ± 12	0.071
Average EE/HR, mL·min^−1^·bpm^−1^	6.38 ± 0.87	7.48 ± 1.32	0.068

Values are mean ± SD. EE, energy expenditure; HR, heart rate; bpm, beats per minute.

**Table 3 ijerph-18-01441-t003:** Changes in body weight and urine variables between pre-trekking and at the hut.

		TW	CE+HCO_3_	Two-Way ANOVA *p* Values		
		(6 M and 2 W)	(6 M and 2 W)	Condition	Time	Interaction
Body weight, kg	Pre	61.66 ± 8.13	63.07 ± 10.24	0.780	<0.001	0.135
	Hut	61.19 ± 7.97	62.40 ± 10.21			
[Na^+^]_u_, meq·L^−1^	Pre	26.0 ± 14.0	29.0 ± 20.0	0.701	<0.001	0.262
	Hut	55.7 ± 21.4	46.4 ± 21.3			
[HCO_3_^−^]_u_, meq·L^−1^	Pre	2.04 ± 0.28	2.23 ± 0.77	0.075	0.020	0.068
	Hut	1.99 ± 0.15	3.36 ± 1.42			
pH, _u_	Pre	7.00 ± 0.76	7.19 ± 1.07	0.956	0.158	0.272
	Hut	6.94 ± 0.62	6.72 ± 0.58			

Values are mean ± SD. Na^+^, sodium ion; HCO_3_^−^, bicarbonate ion.

## Data Availability

All available data are included in this manuscript.

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
