# Peer review of "Effect of Carbohydrate-Electrolyte Solution Including Bicarbonate Ion Ad Libitum Ingestion on Urine Bicarbonate Retention during Mountain Trekking: A Randomized, Controlled Pilot Study"

_ijerph, 2021, doi:10.3390/ijerph18041441_

Round 1
Reviewer 1 Report
Interesting experiment, well designed. I would like to pay attention to
weak points, in my opinion:
Methodology:
- small group size
- predominance of men, analysis of results common for both genders. Unfortunately, due to the very small group size, it is not possible to conduct a separate analysis in women and men. Women's physiology differs from men's physiology, especially during extreme exercise.
- The fluids consumed by the respondents differed not only in the content of buffers (CE + HCO3) but also in calories. In the TW group - zero calories in the CE + HCO3 group - 10 kcal / 100 ml. Better caloric supply affects the incidence of acute mountain sickness (AMS) symptoms - requires a comment
- lack of data on previous episodes of AMS in the studied persons - requires a comment
- references, 20% of publications are older than 20 years - requires changes and updating
Author Response
TO THE REVIEWER #1
Comment: Interesting experiment, well designed. I would like to pay attention to weak points, in my opinion:
Response: We wish to express our appreciation for your insightful comments, which we believe have significantly helped to improve the presentation of our work. Your comments are written in Italic fonts, and our responses are written in normal fonts. Modified sections of the manuscript are written in Red fonts, which we also present here in the response letter.
Comment: Methodology; small group size
Response: As we acknowledge this, and thus, we conducted post-hoc power analysis and obtained sufficient power > 0.95 (page 11, line351-352). However, other reviewers also pointed out sample size calculation. Thus, in the statistic section, we added the description about sample size calculation based on a previous study. Please note that our main outcome is urine HCO3-, and our hypothesis is that urine HCO3- retention increases greater at high altitude with CE+HCO3 beverage ingestion; however, to the best of our knowledge, no studies have tested this hypothesis. Therefore, we used the data which shows the significant difference in urine HCO3- between sea level and high altitude (ref. # 29). Accordingly, we added following descriptions. Note that ref #29 was newly added.
Page 6 Line 202-208
Sample size
A sample size estimation for the primary analysis (urine HCO3-) indicated that approximately 6 participants for each group were needed to produce an 80% chance of obtaining statistical significance at the 0.05 level (G Power 3.1). This required sample size was calculated based on a previous study that a minimum significant change of urine HCO3- between sea level and high altitude > 3,000 m is ~5 mmol L-1 [29]. Eight participants for each group completed the experimental protocol to account for potential missing data.
- Zouboules, S. M.; Lafave, H. C.; O'Halloran, K. D.; Brutsaert, T. D.; Nysten, H. E.; Nysten, C. E.; Steinback, C. D.; Sherpa, M. T.; Day, T. A., Renal reactivity: acid-base compensation during incremental ascent to high altitude. The Journal of physiology 2018, 596, (24), 6191-6203.
Comment: Methodology; predominance of men, analysis of results common for both genders. Unfortunately, due to the very small group size, it is not possible to conduct a separate analysis in women and men. Women's physiology differs from men's physiology, especially during extreme exercise.
Response: We agree this opinion. We added this limitation as below. Note that ref #48 and 49 were newly added.
Page 11 Line 354-358
Similarly, the analysis of results was combined with both men and women. Because of very small sample size (six men and two women for each group), it was impossible to conduct a separate analysis in women and men. However, some previous studies demonstrated that cardiorespiratory response at high altitude was different between men and women [48, 49].
- Caravita, S.; Faini, A.; Lombardi, C.; Valentini, M.; Gregorini, F.; Rossi, J.; Meriggi, P.; Di Rienzo, M.; Bilo, G.; Agostoni, P.; Parati, G., Sex and acetazolamide effects on chemoreflex and periodic breathing during sleep at altitude. Chest 2015, 147, (1), 120-131.
- Takase, K.; Nishiyasu, T.; Asano, K., Modulating effects of the menstrual cycle on cardiorespiratory responses to exercise under acute hypobaric hypoxia. Jpn J Physiol 2002, 52, (6), 553-60.
Comment: Methodology; The fluids consumed by the respondents differed not only in the content of buffers (CE + HCO3) but also in calories. In the TW group - zero calories in the CE + HCO3 group - 10 kcal / 100 ml. Better caloric supply affects the incidence of acute mountain sickness (AMS) symptoms - requires a comment
Response: We appreciate your important suggestion. We cited one more ref. and also added the statement about potential effect of different calorie intake between the groups in the discussion. Please see below. Note that ref #47 was newly added.
Page 10-11 Line 334-337
Another explanation may relate to different calorie intake between the two groups as a previous study reported that reduced energy intake after rapid ascent to high altitude is associated with AMS severity [47]. These diet effects on AMS should also be expanded in the future study.
- Aeberli, I.; Erb, A.; Spliethoff, K.; Meier, D.; Gotze, O.; Fruhauf, H.; Fox, M.; Finlayson, G. S.; Gassmann, M.; Berneis, K.; Maggiorini, M.; Langhans, W.; Lutz, T. A., Disturbed eating at high altitude: influence of food preferences, acute mountain sickness and satiation hormones. Eur J Nutr 2013, 52, (2), 625-35.
Comment: Methodology; lack of data on previous episodes of AMS in the studied persons - requires a comment
Response: In this study, no participants are professional mountaineer with less experience at high-altitude. Therefore, no prior history of AMS. We added this information in the Method section as follows;
Page 3 Line 102-104
Additionally, none of the participants was exposed to an altitude higher than 1,500m within 6 months before the study, and they were confirmed to have without a prior history of symptoms of AMS [23]
Added reference;
[23] Horiuchi, M.; Endo, J.; Dobashi, S.; Kiuchi, M.; Koyama, K.; Subudhi, A. W., Effect of progressive normobaric hypoxia on dynamic cerebral autoregulation. Experimental physiology 2016, 101, (10), 1276-1284.
Comment: Methodology; references, 20% of publications are older than 20 years - requires changes and updating
Response: At first, we cited some original references, i.e., old refs. But according to your suggestions, we replaced old to new and deletes some old references. As a result, we cited 49 references in total, of them, 4 refs. are older than 20 years (ref # = 19, 31, 34, and 39, rate = 8.2%).

Reviewer 2 Report
STRUCTURE
The manuscript is properly structured.
TITLE AND ABSTRACT
The title or abstract should inform that the type of study.
INTRODUCTION
- Line 44: why not all?
- Do not use the first person in scientific publications. Applicable to the entire document (Lines 70, 79…)
MATERIAL AND METHODS
- Describe trial design (such as parallel, factorial) including allocation ratio
- Were there any important changes to methods after trial commencement? Why?
- What were the eligibility criteria for participants?
- Add setting and locations where the data were collected
- How sample size was determined?
- Line 108: what is a light lunch?
- Line 116: was food and beverage ad libitum measured?
- What was the method used to generate the random allocation sequence?
- What was the type of randomisation?
- Who generated the random allocation sequence, who enrolled participants, and who assigned participants to interventions?
- Were participants, care providers or researchers blinded?
RESULTS
- Consider use of a flow diagram of participants
- Include dates defining the periods of recruitment and follow-up
- Why the trial ended?
- A table showing baseline demographic and clinical characteristics for each group is missing
- Line 230: what is the meaning of LLQ?
- Was there any significant damage or unintended effects on each group?
DISCUSSION
- Add generalisability (external validity, applicability) of the trial findings
REFERENCES
- References follow the style indicated.
Author Response
TO THE REVIEWER #2
Comment: STRUCTURE; The manuscript is properly structured.
Response: We wish to express our appreciation for your insightful comments, which we believe have significantly helped to improve the presentation of our work. Your comments are written in Italic fonts, and our responses are written in normal fonts. Modified sections of the manuscript are written in Red fonts, which we also present here in the response letter. Please note that we responded with one answer in response to several comments as they had almost the same meanings.
Comment: TITLE AND ABSTRACT; The title or abstract should inform that the type of study.
Response: We added a following term in the title and abstract.
Title: Effect of carbohydrate-electrolyte solution including bicarbonate ion ad libitum ingestion on urine bicarbonate retention during mountain trekking: A randomized, controlled pilot study
Abstract
Page 1 Line 14-17
This study was a randomized, controlled pilot study. Sixteen healthy lowlander adults were divided into two groups (six males and two females for each): a tap water (TW) group (0 kcal with no energy) and a CE+HCO3 group. The allocation to TW or CE+HCO3 was double blind.
Comment: INTRODUCT ION; Line 44: why not all?
Response: We explained more detail as below.
Page 2 Line 46-47
but a few studies reported no effect of HCO3- ingestion on exercise performance in hypoxia [14, 15].
Comment: Do not use the first person in scientific publications. Applicable to the entire document (Lines 70, 79…)
Response: We changed not to use the first person as much as possible throughout the manuscript. We appreciate that you could look into the manuscript with red fonts.
Comment: MATERIAL AND METHODS
Describe trial design (such as parallel, factorial) including allocation ratio.
Comment: What was the method used to generate the random allocation sequence?
Comment: What was the type of randomization?
Comment: Who generated the random allocation sequence, who enrolled participants, and who assigned participants to interventions?
Response: We added a flow chart figure (new figure 1) with respect to enrollment, allocation etc. of the participants. Further, in the main text, we described additionally as follows;
Page 3 Line 97-109
The participants were recruited for the present study through an advertisement at the Mount Fuji Research Institute and local area of Fuji-yoshida city and Kawaguchiko town. The recruitments were performed from April 8 to June 26, 2016 (n= 29). As shown in Figure 1, applicants who were free from any known known cardiovascular or cerebrovascular diseases and had not taken any medications, were enrolled (n = 16). Additionally, none of the participants was exposed to an altitude higher than 1,500m within 6 months before the study, and they were confirmed to have without a prior history of symptoms of AMS [23]. The allocation to TW or CE+HCO3 was double-blind; experimenters learned of participant’s allocation only after the statistical analysis has been performed on all outcome measures. This random-ized allocation was conducted by a staff-member not directly involved with mountain trekking study, and was stratified by sex and age. The allocation ratio to each group was 1:1 (n =8 for each group).
Comment: Were there any important changes to methods after trial commencement? Why?
Response: One change from the first plan was to finish the study at the hut. This is because very bad weather (i.e., typhoon was coming, so we had to give up further ascend). Additionally, during descending, of course, we had to prioritize safe descending for all participants rather than measurement. We add a simple reason in the results section as below.
Page 6-7 Line 221-224
All of the 16 participants (eight for each group) successfully completed mountain trekking, and there was no missing data. Therefore, all data were used for further analysis. Due to very bad weather condition (i.e., rainy and windy) after reaching the hut, the study was terminated after overnight staying at the hut.
Comment: What were the eligibility criteria for participants?
Response: As aforementioned, prior history of AMS, taking medications, engaging in regular physical activity (professional sports), with cardiorespiratory, and cerebrovascular diseases, were excluded during the process of enrollment. We added some concrete explanations in the main text as well as adding the new figure 1.
Page 3 Line 100-104
As shown in Figure 1, applicants who were free from any known known cardiovascular or cerebrovascular diseases and had not taken any medications, were enrolled (n = 16). Additionally, none of the participants was exposed to an altitude higher than 1,500m within 6 months before the study, and they were confirmed to have without a prior history of symptoms of AMS [23].
Comment: Add setting and locations where the data were collected
Response: We added further explanations as below.
Page 2 Line 83-86
In this study, two huts were provided for pre- (2,305 m) and post- (3,000 m) measurements of mountain trekking. The measurements were performed in a sufficient space at the living room, and the room temperature was controlled at approximately ~22°C.
Comment: How sample size was determined?
Response: As we acknowledge this, and thus, we conducted post-hoc power analysis and obtained sufficient power > 0.95 (page 11, line351-352). However, other reviewers also pointed out sample size calculation. Thus, in the statistic section, we added the description about sample size calculation based on a previous study. Please note that our main outcome is urine HCO3-, and our hypothesis is that urine HCO3- retention increases greater at high altitude with CE+HCO3 beverage ingestion; however, to the best of our knowledge, no studies have tested this hypothesis. Therefore, we used the data which shows the significant difference in urine HCO3- between sea level and high altitude. Accordingly, we added following descriptions. Note that ref #28 was newly added.
Page 6 Line 202-208
Sample size
A sample size estimation for the primary analysis (urine HCO3-) indicated that approximately 6 participants for each group were needed to produce an 80% chance of obtaining statistical significance at the 0.05 level (G Power 3.1). This required sample size was calculated based on a previous study that a minimum significant change of urine HCO3- between sea level and high altitude > 3,000 m is ~5 mmol L-1 [29]. Eight participants for each group completed the experimental protocol to account for potential missing data.
- Zouboules, S. M.; Lafave, H. C.; O'Halloran, K. D.; Brutsaert, T. D.; Nysten, H. E.; Nysten, C. E.; Steinback, C. D.; Sherpa, M. T.; Day, T. A., Renal reactivity: acid-base compensation during incremental ascent to high altitude. The Journal of physiology 2018, 596, (24), 6191-6203.
Comment: Line 108: what is a light lunch?
Response: We apologize for lack of information, and added information as below.
Page 4 Line 129
The participants came to the parking area at 2,305m by car at approximately 13:30 after a light lunch (provided same sandwich) about 1 h prior.
Comment: Line 116: was food and beverage ad libitum measured?
Response: No, we could not check because participants climbed individually at their own speed.
We add an explanation in the discussion as follows;
Page 10 Line 307-310
However, it should also be note that it was not recorded when and how each individual consumed energy food and/or beverage during ascending, suggesting that optimal strategy of bicarbonate ion ingestion during mountain trekking at high altitude is still unclear.
Comment: Were participants, care providers or researchers blinded?
Response: Yes, we added detailed explanations as below.
Page 4 Line 121-124
Notably, the participants were only provided either TW or CE+HCO3 beverage, and nutrient composition and the main aim (i.e., effects of different drinks on physiological responses and symptoms of AMS) of this study were not revealed to the participants. Thus, the participants could know only their own taste [24].
Comment: RESULTS; Consider use of a flow diagram of participants
Comment: Include dates defining the periods of recruitment and follow-up
Response: We made new figure 1 as a timeline (flow) of this study. Please see the new figure 1.
Comment: Why the trial ended?
Response: As know, the altitude of top of Mount Fuji is 3,776 m, but, on the study day, weather was very bad due to typhoon. So, we added as follow. Further, the main aim of this study was to examine effect of different beverage on HCO3 retention and symptoms of AMS at high altitude. So, we evaluated variables only during ascend.
Page 7 Line 223-224
Due to very bad weather condition (i.e., rainy and windy) after reaching the hut, the study was terminated after overnight staying at the hut.
Comment: A table showing baseline demographic and clinical characteristics for each group is missing
Response: We added following variables that we could measure; heart rate (HR) at rest and peak during walking, SpO2 at rest. As all subjects are healthy and had no medications, we did not add other clinical characteristics in this table.
Comment: Line 230: what is the meaning of LLQ?
Response: We apologize it. This is a typo and complete our mistake. We tried to abbreviate symptoms of AMS using Lake Louise Questionnaire Scoring System as LLQ etc. But, we decided not to use the abbreviation as only a few time using. Please see below.
Page 9 Line 265-268
Acute mountain sickness was detected in two of eight participants with TW, defined as “Lake Louise Acute Mountain Sickness Score” of 3 or higher with headache, whereas no AMS participants with CE+HCO3. The range of “Lake Louise Acute Mountain Sickness Score” in the TW were 7, 3, and 2 (n = 1, respectively), 1 (n = 3), and 0 (n = 2).
Comment: Was there any significant damage or unintended effects on each group?
Response: No participants had damaged (e.g., fall-induced injury) except symptoms of AMS. To clarify this, we added following descriptions at the first parts in the Results section.
Page 6 Line 221-222
All of the 16 participants (eight for each group) successfully completed mountain trekking, and there was no missing data. Therefore, all data were used for further analysis.
Comment: DISCUSSION; Add generalizability (external validity, applicability) of the trial findings
Response: We added some practical implications as perspectives before Conclusion. Please see below and the main text.
Page 15 Line 363-369
Perspectives
Given that greater dose of HCO3- may cause acute gastrointestinal distress [17] and bicarbonate buffering system may be more required under hypoxic exercise rather than normoxic exercise [3, 4], our results may have an important practical advice for future field investigations in this research area. The benefits of our results will be applied to not only mountain trekking, but also other sports activities at high altitude (e.g., ball games, running, and cross-country skiing).
Comment: REFERENCES; References follow the style indicated.
Response: Based on all reviewer’s suggestion, we had to replace or add some citations.

Reviewer 3 Report
The manuscript by Masahiro Horiuchi and colleagues well demonstrates the effect of carbohydrate-electrolyte solution including bicarbonate ion ad libitum ingestion on urine bicarbonate retention during mountain trekking. This manuscript is novel and well organized. However, I just have one minor concern.
In the statistical analysis section, please add the power analysis and sample size calculation.
Author Response
TO THE REVIEWER 3
Comment: The manuscript by Masahiro Horiuchi and colleagues well demonstrates the effect of carbohydrate-electrolyte solution including bicarbonate ion ad libitum ingestion on urine bicarbonate retention during mountain trekking. This manuscript is novel and well organized. However, I just have one minor concern.
Response: We wish to express our appreciation for your insightful comments, which we believe have significantly helped to improve the presentation of our work. Your comments are written in Italic fonts, and our responses are written in normal fonts. Modified sections of the manuscript are written in Red fonts, which we also present here in the response letter. Please note that we responded with one answer in response to several comments as they had almost the same meanings.
Comment: In the statistical analysis section, please add the power analysis and sample size calculation.
Response: As we acknowledge this, and thus, we conducted post-hoc power analysis and obtained sufficient power > 0.95 (page 11, line351-352). However, other reviewers also pointed out sample size calculation. Thus, in the statistic section, we added the description about sample size calculation based on a previous study. Please note that our main outcome is urine HCO3-, and our hypothesis is that urine HCO3- retention increases greater at high altitude with CE+HCO3 beverage ingestion; however, to the best of our knowledge, no studies have tested this hypothesis. Therefore, we used the data which shows the significant difference in urine HCO3- between sea level and high altitude. Accordingly, we added following descriptions. Note that ref #28 was newly added.
Page # Line #
Sample size
A sample size estimation for the primary analysis (urine HCO3-) indicated that approximately 6 participants for each group were needed to produce an 80% chance of obtaining statistical significance at the 0.05 level (G Power 3.1). This required sample size was calculated based on a previous study that a minimum significant change of urine HC O3- between sea level and high altitude > 3,000 m is ~5 mmol L-1 [29]. Eight participants for each group completed the experimental protocol to account for potential missing data.
- Zouboules, S. M.; Lafave, H. C.; O'Halloran, K. D.; Brutsaert, T. D.; Nysten, H. E.; Nysten, C. E.; Steinback, C. D.; Sherpa, M. T.; Day, T. A., Renal reactivity: acid-base compensation during incremental ascent to high altitude. The Journal of physiology 2018, 596, (24), 6191-6203.

Round 2
Reviewer 1 Report
The authors significantly revised the original manuscript. They responded to the comments. First of all, they changed the rank of the publication to "pilot study", so their conclusions unfortunately lost their strength. I still believe that the study has its own limitations (group size, no possibility of separate analysis of the results of women and men). However, I believe that an interesting approach to the problem is worth publishing.
Reviewer 2 Report
No further comments.